# Weighted Signed Networks Reveal Interactions between US Foreign Exchange Rates

**DOI:** 10.3390/e26020161

**Published:** 2024-02-12

**Authors:** Leixin Yang, Haiying Wang, Changgui Gu, Huijie Yang

**Affiliations:** Business School, University of Shanghai for Science and Technology, Shanghai 200093, China; 212150983@st.usst.edu.cn (L.Y.); gu_changgui@163.com (C.G.); hjyang@usst.edu.cn (H.Y.)

**Keywords:** detrended cross-correlation analysis, weighted signed network, exchange rate, balanced triads, fluctuation propagation

## Abstract

Correlations between exchange rates are valuable for illuminating the dynamics of international trade and the financial dynamics of countries. This paper explores the changing interactions of the US foreign exchange market based on detrended cross-correlation analysis. First, we propose an objective way to choose a time scale parameter appropriate for comparing different samples by maximizing the summed magnitude of all DCCA coefficients. We then build weighted signed networks under this optimized time scale, which can clearly display the complex relationships between different exchange rates. Our study shows negative cross-correlations have become pyramidally rare in the past three decades. Both the number and strength of positive cross-correlations have grown, paralleling the increase in global interconnectivity. The balanced strong triads are identified subsequently after the network centrality analysis. Generally, while the strong development links revealed by foreign exchange have begun to spread to Asia since 2010, Europe is still the center of world finance, with the euro and Danish krone consistently maintaining the closest balanced development relationship. Finally, we propose a fluctuation propagation algorithm to investigate the propagation pattern of fluctuations in the inferred exchange rate networks. The results show that, over time, fluctuation propagation patterns have become simpler and more predictable.

## 1. Introduction

At present, the world is in the era of globalized finance, and behind every small political change, a series of economic operations are affected. Investors are always concerned about the risk indicators of the market, and the policymakers’ slight improper control of information will trigger a series of chain reactions that cause economic tremors, such as capital flight and exchange rate decline. Changes in economic conditions will again trigger policy changes, and the effects of such operations could again radiate through global financial markets. As an important part of the global economy, the connectedness of the currency markets is now far from a simple connection of numbers and transactions, but has broad implications at the national, industrial, and even individual levels [1]. Due to increasing international trade and cross-border investment, the connectedness of currency system is directly related to the functioning of global commerce. Exchange rate fluctuations have a significant impact on the prices of goods and services and on international capital flows. Moreover, the interconnectedness of macroeconomic policies among countries is reflected in currency connectivity, and adjustments in one country’s monetary policy can affect inflation, interest rates, and employment in other countries. In recent decades, the emergence of financial crises, such as the Mexican financial crisis, the Asian financial crisis, and the subprime crisis in the United States, as well as the economic recession caused by the 9.11 incident, SARS, and the COVID-19 pandemic, have further highlighted the interdependence between currencies [2]. The spread of global financial risks has made transnational cooperation a key means to maintain global economic stability.

Under an increasingly complex global network of interconnected finance, understanding monetary connectivity in financial markets is crucial for economic actors, including investors, economic authorities, and policymakers [3]. This paper explores how the cross-correlation of the exchange rate of the US dollar to a range of currencies changes in different periods. Mastering the exchange rate information that has an important influence on the forex exchange market in different periods is helpful to better cope with economic challenges and reduce the improper or untimely response caused by information bias.

The detrended cross-correlation analysis (DCCA) method is very effective in the measurement of non-stationary sequence relationships and has been widely used in many fields [4,5,6,7,8]. It is an extension of the detrended fluctuation analysis (DFA) method [9]. In subsequent research and development, the DCCA method has been proven to be more accurate and clear than the Pearson correlation coefficient method in identifying nonlinear interactions between time series [10]. Kristoufek [11] made a detailed comparison between the DCCA coefficient and Pearson coefficient from the perspective of the degree of non-stationarity and the setting of important influencing parameters in the equation and proved that the DCCA method has significant advantages in the measurement of non-stationary series cross-correlation. The application scope of DFA and DCCA methods has been extended to the multifractal dimension. Multifractal detrended fluctuation analysis (MFDFA) [12] can detect whether there is a multifractal phenomenon in the time series, and it has been proven that there is a certain inverse relationship between the efficiency of the financial market and multifractality [13,14,15]. The multifractal detrended cross-correlation analysis (MFDCCA) [16] is used to demonstrate multifractal behavior in power–law cross-correlations between series, and there are numerous examples of its effective application, e.g., ref. [17,18,19]. Although the DCCA method can describe the cross-correlation between two non-stationary series, it cannot clearly quantify the strength of the correlation. The DCCA coefficient ($\sigma_{DCCA}$) proposed by Zebende [20] in 2011 solved this problem. In the same year, Podobnik and Stanley et al. [21] tested the statistical significance of the correlation in the range of $\sigma_{DCCA}$. Zebende had also produced some achievements in DCCA research and expansion: in 2013, Zebende and Silva et al. [22] proposed a differential DCCA correlation $\sigma_{DCCA}$(n), which showed a good relationship between αDFA (long-term autocorrelation index) and λDCCA (long-term cross-correlation index). In 2018, Zebende and Filho [23] proposed a new detrended multiple cross-correlation coefficient to quantify the correlation between multiple variables and a target variable [24]. Guedes and Zebende [25] used the sliding window method to transform $\sigma_{DCCA}$ into a time-dependent cross-correlation function in 2019.

DCCA also has many brilliant applications in combination with other strategies, including complex network methods. With the vigorous development of network science, researchers are increasingly illuminating the real world with complex network structure characteristics [26,27,28]. If one financial indicator is regarded as a network node, then the correlation coefficient between indicators can be used to quantify the relationship between different nodes. Combining these pairwise relationships, we can build the network of financial markets, which reflects the global relationship between all considered elements. Shin et al. [29] built a global network of 24 stock markets using the DCCA method, and detected the signal of the financial crisis by observing community structure with the Girvan–Newman method. Wang et al. [30] selected 462 component stocks in the S&P 500 index as research objects, established the empirical cross-correlation matrix of the Pearson correlation coefficient and the detrended cross-correlation number combined with the random matrix theory method, and finally found the characteristics conducive to risk management and optimal portfolio selection in the American stock market on different time scales. Ferreira et al. [31] studied the cryptocurrency market using detrended cross-correlation and detrended moving average cross-correlation correlation coefficients and found a long-term correlation between the returns of cryptocurrencies and their lagging returns of about 30 days. Adam et al. [32] first used ensemble empirical mode decomposition to decompose the daily exchange rates of 15 Southern African Development Community exchange rate markets. Then, they used detrended cross-correlation analysis to analyze the cross-correlation between different frequencies, residuals, and original sequences and constructed a correlation network to obtain richer information.

The $\sigma_{DCCA}$ takes values in the range between $-1\leq \sigma_{DCCA}\leq 1$ [20], and the value’s sign declares the feature of the power–law cross-correlation relationship between series. This suggests the formalism of the signed network, which extends typical networks by accommodating negative as well as positive links [33,34]. The signed network uses positive and negative edges to disambiguate the friendship and hostility between persons in the social network so as to show the interactions between different nodes precisely [35]. Structurally stable and structurally unstable components of a network can be extracted using balance theory and status theory, which can be leveraged to encourage groups to develop strong and reliable configurations [33,36,37,38]. Inspired by this, here we consider using the $\sigma_{DCCA}$ to build signed networks that reveal the changing patterns, balance, and stability in the US foreign exchange market. This allows for a more meticulous and in-depth analysis of exchange rate market interactions, which has rarely been seen in previous studies that have used the network to analyze the forex market.

In this paper, we consider the exchange rate return series of 21 currencies exchanged for US dollars. After removing the seasonal effect of time series, we use the detrended cross-correlation analysis coefficient to quantify the cross-correlation between exchange rates. Then, we use a reasonable method to find the most representative time scale objectively and construct the signed network over different periods. Compared with the non-signed network, the signed network can more clearly show the interconnection relationship between exchange rates. As time goes from far to near, the change in network connection status is a direct reflection of the connectedness of currency markets. After that, we analyze the node centrality of the network in different development periods, and use balance theory to discuss the important node combinations with long-term stable interconnection relationships in the network. The nodes with high network centrality indicate the most influential exchange rate, and the changes in these nodes are likely to drive the changes in the nodes connected with them. The exchange rates represented by the stable node combinations have the most consistent change rule. Finally, we simulate the effect of a sudden change in real exchange rate returns on the entire US foreign exchange market.

In reality, the sudden change in an exchange rate may cause turbulence in the entire foreign exchange market, and the trading risks of the foreign exchange market may radiate outward, which may also spread to the financial markets such as stocks and securities. Adverse developments in financial markets often lead to concern for policymakers because investors have the opportunity to react immediately, but policy intervention takes time. Our findings provide useful theoretical support and practical guidance for enterprises, institutional investors, and policymakers, and provide certain predictive information for the overall change trend of the foreign exchange market in the future, helping them to make economic judgments. The concepts and methods used in data processing and network construction are given in Section 2 below. Section 3 analyzes and discusses the research results in detail. The final section concludes the whole paper.

## 2. Methods

### 2.1. Data and Preprocessing

Daily closing price data of exchange rates for 30 years from 1990 to 2019 were collected from investing.com for the World Bank 2021 top 50 countries by global GDP. Ten of these countries use the euro as their legal tender. In particular, the euro series refers to the European Currency Unit (ECU, for short, until 31 December 1998) and the euro (from 1 January 1999). When the euro was introduced, it replaced the ECU with a 1:1 ratio. The ECU is a unit of account adopted by the European Economic Community, and its value is based on a weighted average of a basket of European currencies. Nineteen exchange rate prices were not recorded continuously in early 1990 and for a period of time after that; they have not been taken into account in the subsequent analysis because long-time continuous missing data are difficult to fill close to the real situation by preprocessing. The specific deficiencies can be seen in Table A1. After this screening, 21 exchange rates of the US dollar against other currencies are finally selected as the research objects. The currencies and corresponding international standard alphabetic codes are shown in Table 1, which refers to the ISO 4217 currency code established by the International Organization for Standardization. In order to make the following analysis and discussion more concise, we use currency alphabetic codes to represent the corresponding exchange rates.

In the process of collecting real data, problems such as omission, misrecording, and repeated recording will inevitably occur. Therefore, missing values and outlier processing of data are often needed to make analysis results more accurate before putting them into the calculation. For the exchange rate data used in this paper, the existing problems and treatment methods are as follows: (i) where data are missing, they will be substituted with the average value of the available data immediately before and after; (ii) for data with multiple recorded values on the day, the last recorded value is used; (iii) non-working day records are not considered. After this data cleaning is completed, the return rate series is calculated according to:(1)$$R_{t}=\ln\frac{P_{t}}{P_{t-1}},$$
where $R_{t}$ and $P_{t}$ are the exchange rate return and closing price, respectively, of day *t*, t=1,2,…,T.

### 2.2. Robust Seasonal-Trend Decomposition

Before the network analysis, we used the robust seasonal-trend decomposition (RSTL) [39] to obtain the residual series of exchange rate closing prices. Removing seasonal effects not only improves the reliability of information but may also help reveal the intrinsic dynamics of exchange rate returns [14,40]. RSTL has robust advantages over other existing decomposition methods in extracting long-term time series trends and dealing with outliers. RSTL considers the classical seasonal-trend decomposition using the locally estimated scatterplot smoothing (LOESS) model [41]: (2)$$y_{t}=T_{t}+s_{t}+r_{t},$$
where t=1,2,…,N, $y_{t}$ indicates a time series of length *N*, $T_{t}$ is the trend of the time series, $s_{t}$ is the seasonal variation in the time series, and $r_{t}$ denotes the remainder. RSTL splits the remainder $r_{t}$ into two parts: (3)$$r_{t}=a_{t}+n_{t},$$
where $a_{t}$ donates the spike or dip, $a_{t}$ is conducive to extracting abnormal change more robustly, and $n_{t}$ denotes white noise.

The RSTL algorithm (the code for RSTL is available at: https://github.com/LeeDoYup/RobustSTL, accessed on 4 July 2023) consists of the following five steps: (1) using bilateral filtering [42] to denoise time series; (2) solving a LAD regression with sparse regularizations to extract trends [43]; (3) using non-local seasonal filtering [39] to calculate seasonal components; (4) adjusting the extracted components; (5) repeating steps (1)–(4) until convergence.

### 2.3. Detrended Cross-Correlation Analysis

Detrended cross-correlation analysis typically requires four calculation steps [4,20]. Consider two non-stationary time series x(1),…,x(N) and y(1),…,y(N). Here, *x* and *y* are equal to the residual series obtained from Section 2.2 above.

Step 1: Integrate time series *x* and *y* as follows for k=1,2,⋯,N: (4)$$\left\{\begin{array}{c} R_{x}\left(k\right)=\sum_{t=1}^{k}x\left(t\right)\\ R_{y}\left(k\right)=\sum_{t=1}^{k}y\left(t\right). \end{array}\right.$$

Step 2: Divide $\left\{R_{x}\left(k\right)\right\}$ and $\left\{R_{y}\left(k\right)\right\}$ into (N−s) overlapping boxes of the same length *s*, where $4\leq s\leq \frac{N}{10}$ in this paper.

Step 3: The covariance of the residuals is then calculated as follows: (5)$$f_{DCCA}^{2}\left(s,i\right)=\frac{1}{\left(s+1\right)}\sum_{k=i}^{i+s}\left(R_{x}\left(k\right)-{\tilde{R}}_{x,i}\right)\left(R_{y}\left(k\right)-{\tilde{R}}_{y,i}\right).$$ In this expression, ${\tilde{R}}_{x,i}$ and ${\tilde{R}}_{y,i}$ are the local linear trends in each box, calculated using least squares.

Step 4: The detrended covariance function is calculated as $F_{DCCA}^{2}\left(s\right)=\frac{1}{\left(N-s\right)}\sum_{i=1}^{N-s}f_{DCCA}^{2}\left(s,i\right)$. When only one time series is analyzed (x=y), the detrended covariance function $F_{DCCA}^{2}\left(s\right)$ turns into a detrended variance function: $F_{DFA,x}\left(s\right)$ and $F_{DFA,y}\left(s\right)$. In the DFA method, the power–law relationship between the $F_{DFA,x}\left(s\right)\left(F_{DFA,y}\left(s\right)\right)$ and *s* of a single non-stationary time series is provided as: $F_{DFA,x}\left(s\right)∼s^{\alpha_{DFA,x}}\left(F_{DFA,y}\left(s\right)∼s^{\alpha_{DFA,y}}\right)$. The DCCA method is designed to explore the power–law cross-correlation between two non-stationary time series with the same length: $F_{DCCA}^{2}\left(s\right)∼s^{\beta_{DCCA}}$. If the power–law cross-correlation appears, then $\beta_{DCCA}\approx \frac{\alpha_{DFA,x}+\alpha_{DFA,y}}{2}$ [4,9]. Because $\beta_{DCCA}$ cannot quantify the level of cross-correlation [20], the DCCA cross-correlation coefficient is defined as follows: (6)$$\sigma_{DCCA}=\frac{F_{DCCA}^{2}\left(s\right)}{F_{DFA,x}\left(s\right)F_{DFA,y}\left(s\right)},$$
where $\sigma_{DCCA}$ quantifies the cross-correlation between two non-stationary series to $\left[-1,1\right]$. Important values of $\sigma_{DCCA}$ include: (7)$$\sigma_{DCCA}=\left\{\begin{array}{cc} 1, & \mathrm{perfect}\;\mathrm{cross}\;\mathrm{correlation};\\ 0, & \mathrm{no}\;\mathrm{cross}\;\mathrm{correlation};\\ -1, & \mathrm{perfect}\;\mathrm{anti}\;\mathrm{cross}\;\mathrm{correlation}. \end{array}\right.$$

### 2.4. Network Centrality

Centrality measures help us find nodes that have essential influence in a network. Here, we list the six centrality definitions we consider in this work. Equations (Equation 8)–(Equation 12) are mature network centrality indexes, and Equation (Equation 13) is a network centrality that we propose.

#### 2.4.1. Degree Centrality

$DC_{i}$ is the normalized degree centrality [44,45] of node *i*:(8)$$DC_{i}=\frac{k_{i}}{n-1},$$
where $k_{i}$ is the degree of node *i* and *n* is the number of nodes.

#### 2.4.2. Betweenness Centrality

$BC_{i}$ is the betweenness centrality [45,46] of node *i*:(9)$$BC_{i}=\frac{2}{\left(n-1)(n-2\right)}\sum_{s,t\neq i}\frac{d_{st}\left(i\right)}{d_{st}},$$
where $d_{st}$ is the number of shortest paths from node *s* to node *t*, while $d_{st}\left(i\right)$ is the number of these shortest paths that pass through node *i*.

#### 2.4.3. Closeness Centrality

$CC_{i}$ is the closeness centrality [45] of node *i*:(10)$$CC_{i}=\frac{n}{\sum_{j=1}^{n}d_{ij}},$$
where $d_{ij}$ is the network distance from node *i* to node *j*, i.e., the number of edges in the shortest path from *i* to *j*. If there is no path reachable between nodes *i* and *j*, so that $d_{ij}$ is infinite, we consider the reciprocal of $d_{ij}$ to be 0.

#### 2.4.4. Eigenvector Centrality

$EC_{i}$ is the eigenvector centrality [47] of node *i*:(11)$$EC_{i}=z_{i}=c\sum_{j=1}^{n}w_{ij}z_{j},$$
where *c* is a proportionality constant and $\left(w_{ij}\right)$ is the weighted adjacency matrix of the network. The $EC_{i}$ depends on both the degree of node *i* and the importance of its neighbor nodes.

#### 2.4.5. Weight-Sum Centrality

$W_{\mathrm{sum}}\left(i\right)$ is the sum of all weights of edges involving node *i*:(12)$$W_{\mathrm{sum}}\left(i\right)=\sum_{j=1}^{n}w_{ij},$$

#### 2.4.6. Neighbor-Sum Centrality

$N_{\mathrm{sum}}\left(i\right)$ emphasizes whether the neighbors of node *i* are influential, defined as the sum of weight-sum $W_{\mathrm{sum}}\left(j\right)$ for all neighbors *j*.
(13)$$N_{\mathrm{sum}}\left(i\right)=\sum_{j\in V(i)}W_{\mathrm{sum}}\left(j\right),$$
where V(i) is the set of nearest neighbors of node *i*. In the weighted adjacency matrix $\left(w_{ij}\right)$, the number of nearest neighbors is equal to the number of non-zero elements in row *i* or column *j* of the node.

## 3. Results and Discussion

We calculate the coefficient values between exchange rates under different time scales: a total of 29 different *s* in the $\left[4,\frac{N}{10}\right]$ interval [11,25,30]. For each time scale *s*, we sum the absolute value of all DCCA coefficients. We find that, for each decade, this sum exhibits a similar pattern (Figure 1), increasing first and then decreasing as the time scale *s* increases. To choose an appropriate time scale $s=s^{*}$ for our analysis, because a higher magnitude coefficient value reflects a more statistically significant relationship, we consider maximizing this sum of magnitudes of DCCA coefficients. As defined above, the greater the absolute value of $\sigma_{DCCA}$, the stronger the interaction between exchange rate returns. To obtain one time scale that could make the sum of DCCA coefficients for three decades all at a higher level, we superimpose the sum of the three periods together to select the most representative time scale for the whole, identifying $s^{*}=53$ as the optimal value (see Figure 1), which leads to a relatively high value of summed DCCA coefficients for each of the three periods. This value is fixed to that used to build networks between exchange rates.

### 3.1. Weighted Signed Network

The $\sigma_{DCCA}$ quantifies the cross-correlation between −1 and 1, and whether this value is positive or negative has important effects. In this paper, the statistical test method proposed by Stanley et al. [21] is used to conduct a confidence test at a 95% level for the coefficients, and we only retain the values where the $\sigma_{DCCA}$ are significant in all time scales. On this basis, we regard each exchange rate as a node, and significant cross-correlation between two exchange rates as corresponding to an edge. High-magnitude cross-correlation means a strong relationship, so we use the cross-correlation coefficient as the edge weight. Since cross-correlation reflects the bi-directional relationship, we build an undirected-weighted signed network. Then, for the three research samples, 1990–1999, 2000–2009, and 2010–2019, we can build three different undirected-weighted signed networks, which can describe both the strength of cross-correlation and whether it is positive or negative.

Figure 2 shows the network structure of the three periods, where orange lines represent the positive edge, while the negative edges are represented as a blue dotted line. For all connected edges, the wider the connected edge, the greater the edge weight. Comparing the three periods, we see that with passing time, the network structure becomes denser, and we also see that negative edges are much lower in number and weight than positive edges. Negative edges only show up in the first two periods, from 1990–1999 to 2000–2009, and the number of these edges decreases. In order to isolate the effects of different strengths of cross-correlation, we separately build Non-weak and Strong networks (Figure A1). Non-weak and Strong networks, respectively, include only edge weights larger than 0.333, i.e., $\sigma_{DCCA}\geq 0.333$, and more than 0.666, i.e., $\sigma_{DCCA}\geq 0.666$ [48]. Isolated nodes are removed from these networks. The original Complete networks have more information, so unless stated otherwise, we focus on these.

From Figure 2a, we can notice that the CNY, COP, and KRW are missing from the network. The reason is that there is no power–law cross-correlation between the changes in these three exchange rates and any other exchange rates or that the $\sigma_{DCCA}$ fails the significance test at one or more time scales [21]. This may be influenced by the 1997 Asian financial crisis [49], which affected Japan, South Korea, China, and other places, resulting in the sharp depreciation of Thailand, South Korea, and other countries [50,51]. At the same time, most of the major Asian stock markets fell sharply, which hindered the development of the Asian economy [52].

Another obvious finding is that there are three negative edges, and all are associated with MXN. This may be related to the Mexican financial crisis in 1994–1995 [53]. In this large-scale crisis, the peso exchange rate fell substantially, stock prices plunged, foreign investors sold pesos frantically to buy dollars, and Mexico’s foreign exchange reserves were significantly reduced [54,55]. This series of economic changes caused market panic and had a profound impact on the economic development of Mexico and even the world [56]. As seen from Figure 2b, MXN also participated in the only negative edge in the subsequent decade.

In the first decade of the 21st century, the pattern of world economic development changed, and the total number of connected edges in the entire network doubled (specific details can be found in Table A2). Not only does the total number of nodes increase (while SAR disappears from the network), but the interaction between nodes also increases (Figure 2b). In 2007, a crisis broke out in the United States due to the subprime mortgage loan problem, which caused severe fluctuations in the stock market with insufficient liquidity in the financial market [57,58,59]. The impact of the subprime crisis eventually spread around the world. In 2009, the GDP and other economic indicators of most countries in the world plummeted. However, for the whole foreign exchange market in the United States, the co-evolution law between the closing prices of different exchange rates may be more consistent.

From 2010 to 2019, the network connections continued to increase, and all of them were positive (Figure 2c). At this time, the interaction of the entire US foreign exchange market is the most complex, and almost every exchange rate has a positive correlation with various other exchange rates. This highlights how contemporary world economic development is trending toward economic globalization [60,61]. The International Monetary Fund pointed out that the increase in the scale and form of transnational trade, capital flow, and the wide and rapid spread of technology have enhanced the interdependence of the world’s economy [62,63].

### 3.2. Network Centrality Ranking

This section analyzes the network centrality of the previously Complete, Non-weak, and Strong networks. We calculate the network centrality ranking for each exchange rate based on Section 2.4. Since weighted signed networks have both positive and negative properties, we define the distance $d_{ij}$ between neighboring nodes *i* and *j* with a formula analogous to that for Pearson distance [64]: $d_{ij}=1-w_{ij}$. For a higher absolute value of a positive (negative) weight $w_{ij}$, the smaller the rectified distance $d_{ij}$ should be. The weight-sum centrality is calculated by $W_{\mathrm{sum}}\left(i\right)$ = $\sum_{j=1}^{m}\left|w_{ij}\right|$, where $\left|w_{ij}\right|$ represents the absolute value of the weight between nodes *i* and *j*. Here, we show the six types of network centrality of exchange rate nodes in networks, which are presented in Figure 3, Figure 4 and Figure 5. Currencies are arranged on the horizontal axis in alphabetical order.

Based on the network centrality ranking results shown in Figure 3, we can see how each exchange rate plays a role in the market and how these roles change with time. Specifically, in the period 1990–1999 (Figure 3a), DKK, EUR  (before 1999, EUR refers to ECU, the same as below), and NOK had high DC and EC. SGD has many weak correlations with other exchange rates; although there are many connected edges, their intensity is low, and that is why SGD disappears from the Strong network (Figure A1d). The order of nodes’ importance varies greatly with the simplification of network structure. NZD, AUD, CAD, and ZAR are the busiest nodes in the Complete network (Figure 3a), while SGD is the hub in the Non-weak network (Figure 4a). The reason behind this is that nodes in Oceania and Asia are connected to nodes in Europe through SGD (Figure A1a). Nodes with high BC can potentially affect information propagation by facilitating, hindering, or even directly altering communication between other nodes [65]. In particular, the importance of NOK, SEK, and CHF differ greatly in BC and CC. In general, the high CC node quickly transmits information through the short circuit radial to others without having to obtain information from further outside. CC increases with the increase in the adjacency of a node. However, the BC decreases as the number of neighbors increases, because this makes the node less likely to be a bridging node. In the centrality of $W_{\mathrm{sum}}\left(i\right)$, the highest nodes in the three networks are nearly the same, which again indicates that nodes such as EUR, DKK, and NOK have a great influence. However, for $N_{\mathrm{sum}}\left(i\right)$, the ranking results show that the nearest neighbors of NZD, ZAR, and GBP have established a relatively significant correlation with other exchange rates. The difference in the impact of MYR and SGD on the network is much greater than their level of cross-correlation. MYR was one of the origins of the 1997–1998 Asian financial crisis [49], but studies have shown that Singapore, also in the region, managed to avoid the worst effects of the crisis and recovered in a short time. There is also evidence that Australia was less strongly affected by the crisis [66,67], and our results are consistent with this conclusion.

During the second study period, the importance of SGD, GBP, and INR in the Complete network showed obvious advantages (Figure 3b), and the ranking of SGD was higher than in the last period. It is interesting to note that INR ranked near the bottom previously (Figure 3a), but the ranking improved significantly in Figure 3b. After the crisis at the end of the last century, India and China became the world’s two fastest-growing major economies, and they are also developing countries with great economic development potential. China’s per capita GDP was about 33% lower than India’s in 1975, but by 2010, it had surpassed India’s by a factor of three [68]. A horizontal comparison of the GDP of the countries involved in the research sample since 1990 shows that China and the United States have maintained a rapid growth pattern in terms of overall GDP growth, much faster than other countries. Therefore, we guess that it may be because of the differences in economic growth rate that the change rule of CNY is not strongly correlated with INR or other exchange rates.

In Figure 4b, the importance of AUD and CAD has become stronger. Although SGD still ranks first among all nodes in terms of BC, AUD has become prominent, which indicates that the interaction between AUD and other exchange rates has become more important. The results of Figure 5b show that EUR and DKK remain the most influential in the Strong network, with EUR having higher values of most centrality measures than others. In this case, the EUR is a transfer center of influence, given the different ways in which nodes occupying higher positions of different centrality indicators transmit information. The importance of CNY, JPY, HKD, and TRY is always non-significant and performs similarly in different networks. The first three exchange rates belong to Asia, while Turkey stretches from Asia to Europe. This reflects the unevenness of the exchange rate trading markets in different developing regions, with the current results showing that Europe is one of the most active regions in the US foreign exchange market. KRW first appeared in the network during 2000–2009 (Figure 2b and Figure 3b), and the results during and after this time show that the influence of KRW in cross-correlation networks continued to increase. In fact, foreign investment is an important part of the Korean financial market, with the foreign investment proportion reaching 40% in the decade after the 1997 Asian financial crisis [69]. In terms of import and export data (relevant data can be found at: https://wits.worldbank.org/, accessed on 12 October 2023), Korea has shown an overall growth trend since 1990 and gradually surpassed Asian countries such as Malaysia, Japan, and Singapore since 2010. The KRW’s increased connectivity with other currencies indicates the growing influence of Korea.

In the period from 2010 to 2019 (Figure 3c, Figure 4c and Figure 5c), with the transformation of network structure from earlier periods, it can be seen that the importance of NOK is increasing, showing that it has greater control over the flow of information over time. At this stage, even though CHF plays an important role in DC and $N_{\mathrm{sum}}\left(i\right)$ of the Complete network, its overall importance drops significantly compared to the previous decade, not only disappearing in the Strong network (Figure 5c) but also ranking last in the Non-weak network (Figure 4c). The JPY has no cross-correlation with other exchange rates greater than or equal to 0.333. In fact, the collapse of the Japanese bubble economy in the late 1980s led to a long period of stagnation and downturn in the Japanese economy [70]. From 1990 to 2000, Japan’s average GDP growth rate was only about 1.28% (the GDP data is available in: https://wits.worldbank.org/, accessed on 12 October 2023), much lower than other developed and emerging countries. The global financial crisis hit the Japanese economy hard again in 2009, and these events impacted the yen’s significance.

The overall ranking of CNY in 2010–2019 increased, especially $N_{\mathrm{sum}}\left(i\right)$, which suggests that it is related to nodes with great influence. For the Strong network (Figure 5), the difference between the first two periods is mainly reflected in the weakening influence of GBP (Figure 5b compared with Figure 5a), while in the third sample period (Figure 5c), CHF and GBP disappear from the network and CAD, KRW, and SGD join in. Brexit, in which Britain exited the European Union, is arguably one of the most important global events in the decade 2010–2019. Brexit has made foreign investors less confident in the development prospects of the British economy, and the pound sterling has depreciated by 11% against other major currencies since the referendum, while the depreciation caused inflation in Britain to rise by about 1.7% [71,72]. The referendum has had an adverse impact on the UK’s economic development, which may also be the reason why GBP nodes suddenly disappeared from the US foreign exchange Strong network.

### 3.3. Balance Theory Analysis

In sociological and psychological theory, relationships between people in social networks are broadly classified into the following four categories [36]: (1) my friend’s friend is my friend; (2) my friend’s friend is my enemy; (3) the enemy of my enemy is my friend; (4) the enemy of my enemy is my enemy. However, at least three nodes can form a signed network with the above characteristics in a complex network. For a signed network with only three nodes, there are also four kinds of connected edge relations (as shown in Figure 6). Balance theory, which plays a critical role in the analysis of signed networks, gives the concept of balanced triads (inside the black dashed line in Figure 6). The relationship between nodes in the balanced triad is more intuitive, stable, and likely to endure [38]. Balanced triads are ideologically consistent with the types in which the number of “enemy” relationships that appear is even. In other words, triangles with an even number of negative sides are balanced triads [36].

Figure 7 shows the balanced strong triads in the three research periods. We take the balanced triad with the largest sum of weights as the balanced strong triad of the Complete network in each period. The strong triad in one research period of different networks is the same. Figure 7a shows the balanced strong triad during sample one, in which the closing prices of the EUR, DKK, and CHF show the closest trend of codirectional movement, showing the best “friend relationship”. DKK, EUR, and CHF maintained a balanced relationship in 2000–2009 (Figure 7b), and their connectedness to each other became stronger. After another decade of development, SEK replaced CHF to form the most stable development relationship with EUR and DKK (see Figure 7c). Comparing the three periods, it can be found that, regardless of the changes in the other exchange rates, EUR and DKK are always “best friends”.

We analyze the geographical distribution of exchange rate markets included in the Strong networks (see Figure 8). Because the ECU was not a circulating currency, we do not consider the geographical distribution in the first sample. The latitude and longitude information involved comes from Google Earth. In Figure A1, the number of nodes included in the Strong networks has not changed between the last decade of the 20th century and the first decade of this century, but more nodes have been added in the decade 2010–2019. As can be seen from Figure 8, CAD is located in North America and KRW and SGD are in Asia, suggesting that the strong cross-correlation between different forex markets is expanding globally, unlike in earlier decades, when a small fraction of nodes were localized to Europe. This conclusion corresponds to the economic globalization we discussed earlier.

Although there is no difference in nodes under the Strong network in the first two decades, and they all contain 10 balanced triads, the balanced triads are not exactly the same (as shown in Table 2, Table 3 and Table 4). According to the weight sum of the balanced triad, we arrange it from high to low. The most prominent difference is that GBP did not belong to any triad during 2000–2009, but it established different equilibrium development relationships with DKK, EUR, and NOK during 1990–1999. Even with this shift in balance, the global efficiency [73] ($GE=\frac{1}{\frac{1}{2}N\left(N-1\right)}\sum_{i>j}\frac{1}{d_{ij}}$) of the Strong network still shows an increasing trend. Our calculations show that the GE in 1990–1999 was 2.8558, and it rose to 6.1057 in 2000–2009. The increase in GE illustrates the improvement in information transmission efficiency in the network. By 2010–2019, the global efficiency of the Strong network had increased again to 7.0282, but at this time, GBP had disappeared in the strong power–law relationship network. The number of nodes in the network had increased, but the number of balanced triads was only four (Table 4), i.e., the exchange rate market with stable “friend” relationships in the network had significantly decreased. This indicates that the exchange market tends to be a non-serial-driven movement, and the strong influence relationship between exchange rates ($\sigma_{DCCA}\geq 0.666$) is in the process of expanding around the world, which is accompanied by volatility.

### 3.4. Propagation on Signed Network

From the perspective of the dynamics of spreading on signed networks [74], we try to explore how the whole US foreign exchange market network will change when a certain exchange rate market suddenly tremors. Algorithm 1 is a simple demonstration of the simulation process. For the network structure of different research periods, we assume that the initial value of all nodes is 0, and if the value of any node increases by 1, the node will impact its neighbor node. The probability of its neighbor node increasing or decreasing by 1 equals the absolute value of the edge weights between them. Whether they increase or decrease depends on whether their cross-correlation is positive or negative. We assume that, after the price of a market changes under the influence of the surrounding environment, it will not change again in a short time, and even if a node is constantly affected by propagation, its value will only change once.
**Algorithm 1** Simulating propagation process**Input:** The complex network graph G(V,E) and initially perturbed node *i*;**Output:** The final state $v_{j}$ of each node *j*; 1:$v_{i}=+1$ 2:U={} 3:**for** node j∈V **do** 4:   $v_{j}=0$; 5:**end for** 6:**while**U≠V**do** 7:   **for** each node j∈V∖U such that $v_{j}\neq 0$ **do** 8:     U=U∪{j}; 9:     **for** each neighbor *k* of each node j∈U such that $v_{k}=0$ **do**10:        randomly generate r∼U[0,1]11:        **if** $\left|w_{jk}\right|\geq r$ **then**12:          **if** $w_{ij}>0$ **then**13:             $v_{k}=v_{k}+1$;14:          **else**15:             $v_{k}=v_{k}-1$;16:          **end if**17:        **end if**18:     **end for**19:   **end for**20:**end while**

Figure 9, Figure 10 and Figure 11 show the Sankey (SankeyMATIC: http://sankeymatic.com/build/, accessed on 14 September 2023) diagram of each node in the three research periods as the final network distribution state when the initial node changes. The left part is the exchange rate chosen for a positive fluctuation (+1), and the right part illustrates the resulting fluctuations of other exchange rates along with the number of times this fluctuation was observed. Each node selected for initial positive fluctuation undergoes 100,000 independent experiments. To ensure the validity of the results, we will not consider final distributions that appear less than 5% of the total times. The number of nodes with −1, 0, and 1 in each distribution is listed separately in parentheses. In the final fluctuation distribution, if only the initial node is non-zero, the single “1” in the array is deleted. This situation occurs because the network is not affected by local tremors. The number of nodes with a final value of “1”, shown in parentheses after all arrays in the graph, is also reduced by 1.

In 1990–1999, there were eight nodes that did not affect the state of other exchange rates in the network when they were initially perturbed, and together they form the largest proportion of an array distribution (Figure 9). Most of the other nodes’ changes will cause the whole network to exhibit a variety of different fluctuation distributions, and their likelihood is not very different. During sample one, when INR is used as the initial change node, the number of valid experiments is the highest, and there are only two final fluctuation distributions. The difference between the two is whether CAD changes under the influence of propagation and INR mutation has no impact on the whole network, accounting for a large majority. In this stage, three negative edges are extended from MXN, its final value has three different cases, and the distribution with a final value of −1 only accounts for $\frac{4}{11}$, while there are six nodes whose value is still 0. As the decades pass, the number of experiments obtained by more than 95% effective statistics increases, but the corresponding number of distributions decreases, which shows that increases and decreases in the forex exchange market network are more and more consistent and stable.

There are only three nodes as initial mutation points that change completely independently of other nodes in the network (Figure 10); during 2010–2019, though, there is only one such node: JPY. However, JPY did not show such a propagation distribution during 1990–1999. This suggests that its position in the US foreign market has become less influential over time. Among the three effective distributions in Figure 11, excepting that we manually increase the value of the JPY node by 1, the difference between the other two distributions is limited to whether the JPY changes. When CHF, CNY, DKK, EUR, GBP, SEK, and SGD serve as initial nodes, the entire network increases by 1 positively, and no other effective distribution appears. The four exchange rates other than CHF, CNY, and GBP are all in the balanced strong triads, suggesting that active influence may spread through robust balanced triangles, and sometimes the indirect effects are stronger.

By comparing Figure 9 to Figure 11, it is found that 0 occurs more frequently in the first decade and least in the last decade of our study sample, which is a result of the increase with time of network density. With the 30-year evolution of the exchange rate market, the exchange rate market has shown overall fragility, and the surge of one exchange rate can easily induce others to change in the same direction. This is consistent with the exchange rate network structure of Section 3.1, where the exchange rate markets show a trend of increasing positive correlation rates. In addition, another important discovery is the role of negative edges, which directly reflect the reverse variation relationship during the propagation process. On the whole, no matter which node’s value changes in the third research stage, the state of the entire network will eventually be limited to three distributions. Compared with Figure 3, the results of Figure 9, Figure 10 and Figure 11 are consistent with network node centrality. Specifically, nodes with high network centrality tend to have a more stable and effective impact during the propagation process, resulting in most nodes ultimately changing their states. These greater numbers of central nodes also correspond to clearer results, revealing positive and opposite fluctuations, as well as the probability of fluctuations.

## 4. Conclusions

We use cross-correlation to examine the global relationships between exchange rates based on network theory. Using the residual series of exchange rate returns, we build weighted signed networks to reflect the positive or negative connections and the intensity of cross-correlation. According to the absolute value of $\sigma_{DCCA}$, the weight of the connected edge is divided into three levels, clearly showing the strength and weakness distribution of the cross-correlation between currency exchange rates. Overall, the negative edges induced by MXN gradually decrease and eventually disappear, the strength and number of positive weights in the network continue to increase, and the edges greater than 0.666 come to involve more nodes.

From the perspective of network centrality, the node importance of INR and JPY has undergone opposite drastic changes around the 21st century, which are strongly related to their countries’ economic development. After the edges less than 0.333 and 0.666 were deleted one after another, many nodes were isolated from the network, and by comparing the line chart, it can be found that the node composition in the Strong network of the third period underwent a large change. CAD, KRW, and SGD enter the high-impact exchange rate network for the first time. The variation in CHF is especially worth studying because it has a strong mutual influence relationship with important foreign exchange markets in the previous period. The BC curve of Figure 5c shows that SGD is an important hub for connecting the entire network structure. The change in the balanced strong triad over time illustrates again that the importance of CHF is decreasing in the US foreign market, while the yield movements of DKK and EUR are becoming more and more consistent. The sharp decline in the number of balanced triads in Table 4 partly reveals the instability in the overall development of the US foreign exchange market, even as more and more exchange rates begin to establish correlations with each other.

We also simulate fluctuations in all exchange rates caused by fluctuations in one single exchange rate. Our method is able to visually depict the impact of each exchange rate fluctuation on all others and the proportion of it that can occur. These propagation simulation results show that the number of network distributions generated when different nodes are positively perturbed is becoming smaller and smaller, and it is based on the increasingly effective experiments for statistics, which may indicate that the imbalance of different exchange rate return development in the US exchange market is weakening.

## Figures and Tables

**Figure 1 entropy-26-00161-f001:**
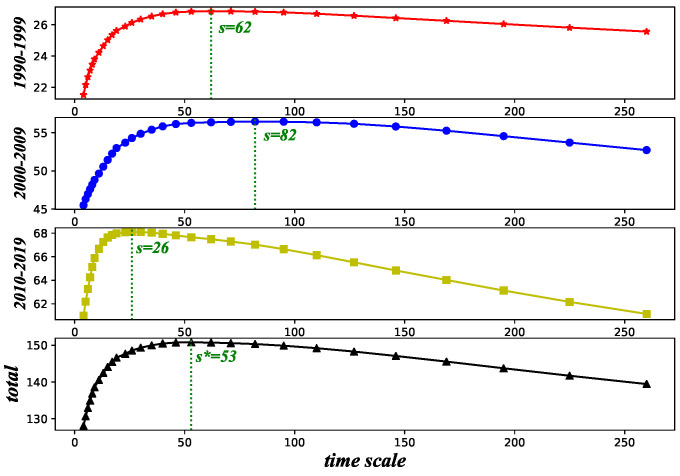
The DCCA coefficients at different time scales.

**Figure 2 entropy-26-00161-f002:**
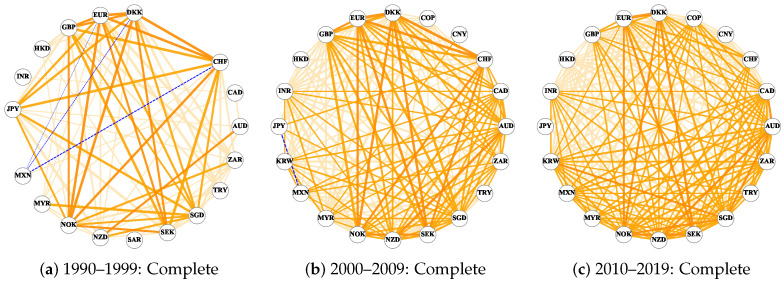
Undirected signed Complete networks in different decades.

**Figure 3 entropy-26-00161-f003:**
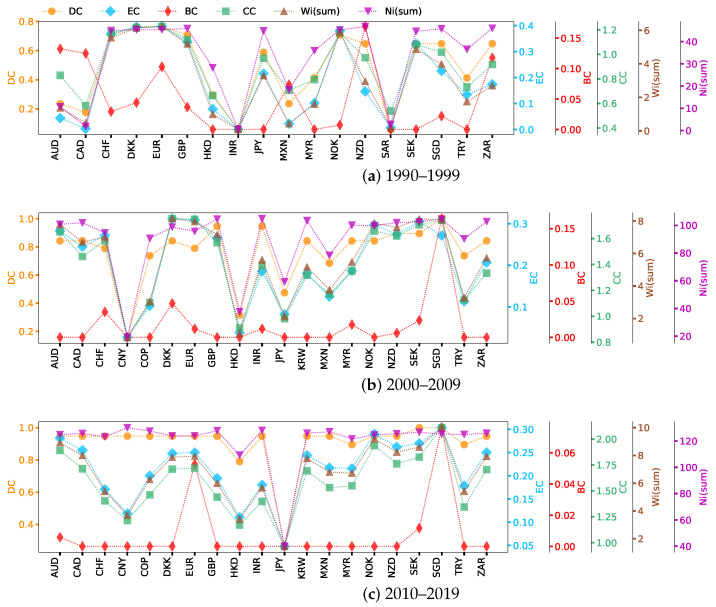
Network centrality for Complete networks over three study sample periods.

**Figure 4 entropy-26-00161-f004:**
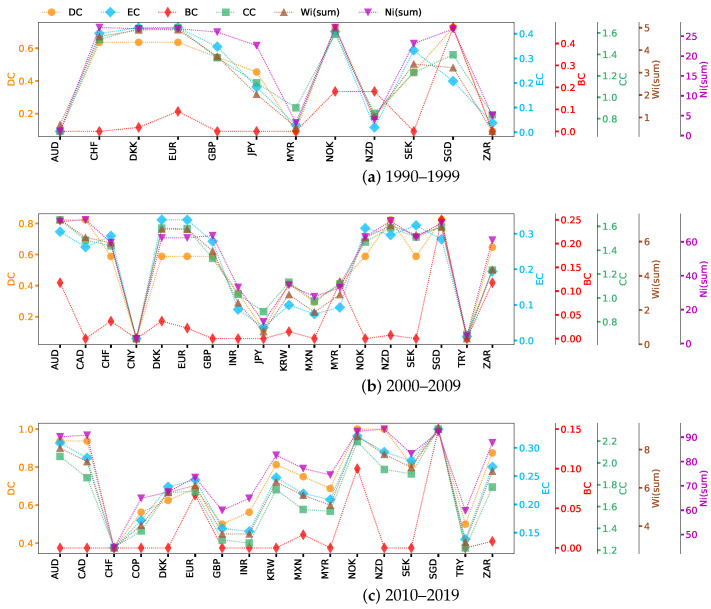
Network centrality for Non-weak networks over three study sample periods.

**Figure 5 entropy-26-00161-f005:**
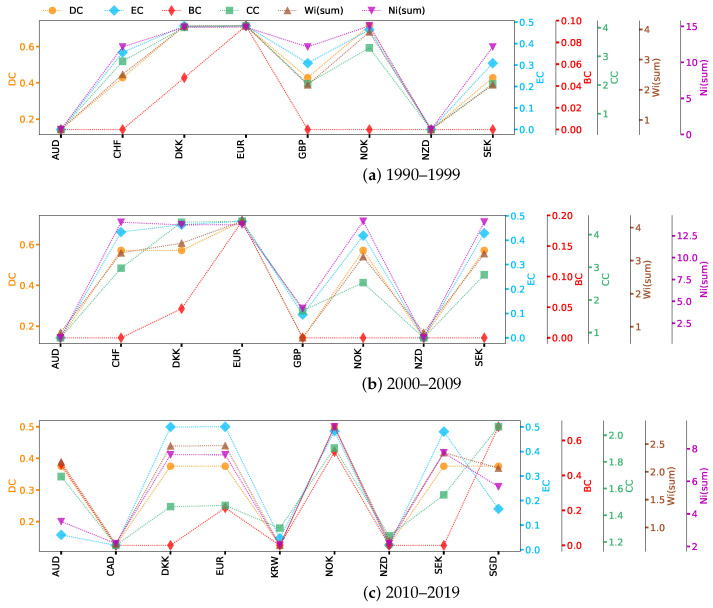
Network centrality for Strong networks over three study sample periods.

**Figure 6 entropy-26-00161-f006:**
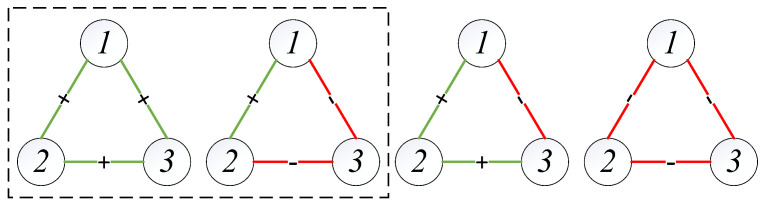
Balance theory.

**Figure 7 entropy-26-00161-f007:**
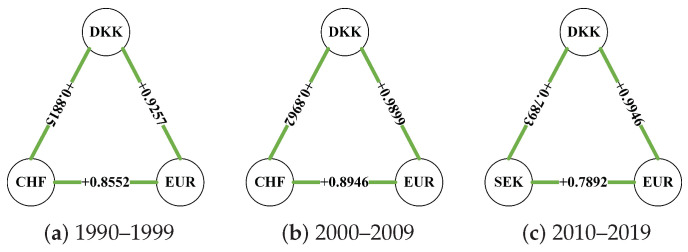
The balanced triangles with the largest sum of weights.

**Figure 8 entropy-26-00161-f008:**
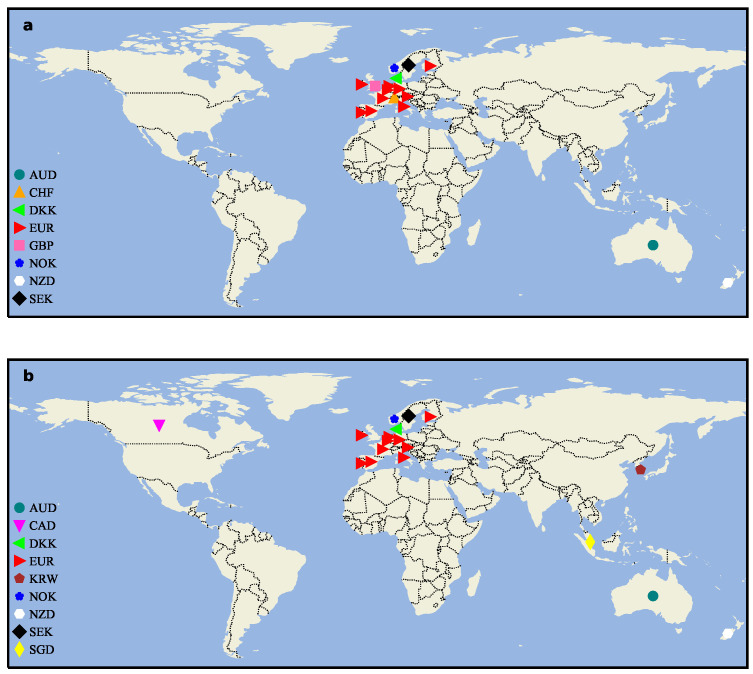
The geographical distribution of exchange rate nodes under the Strong networks. (**a**) The nodes geographical distribution of the second research sample (2000–2009). (**b**) The nodes geographical distribution of 2010–2019. Note: Spain (formerly, the peseta) and the Netherlands (formerly, the Dutch guilder) have used the euro as official currency since 2002.

**Figure 9 entropy-26-00161-f009:**
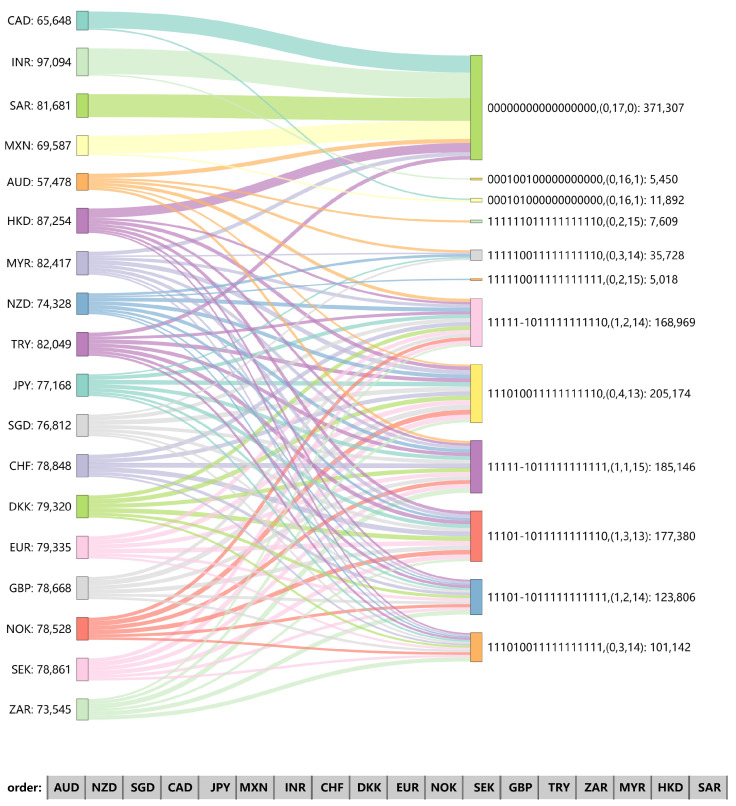
Distribution of final node values in the signed network under different initial node changes during 1990–1999.

**Figure 10 entropy-26-00161-f010:**
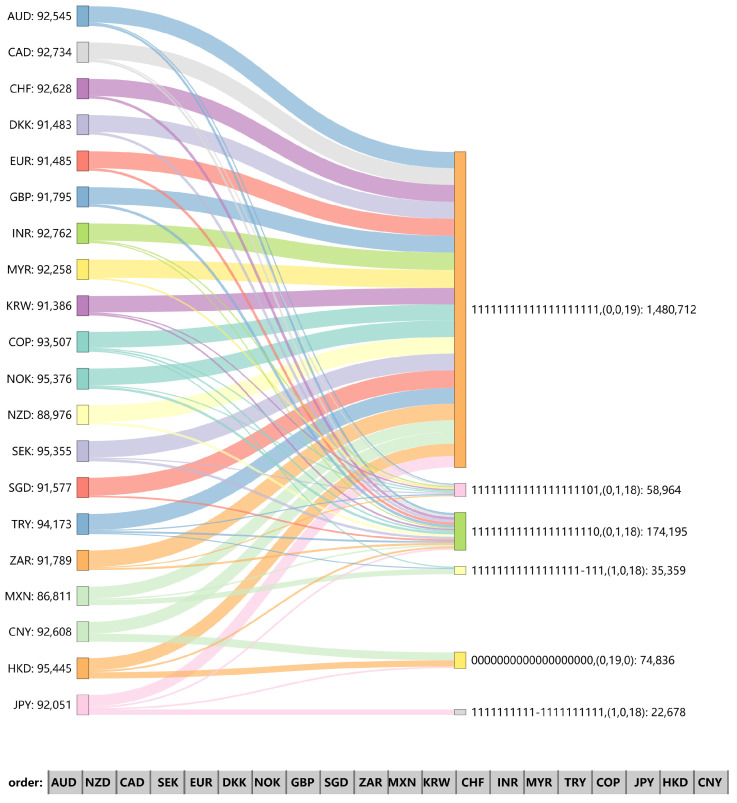
Distribution of final node values in the signed network under different initial node changes during 2000–2009.

**Figure 11 entropy-26-00161-f011:**
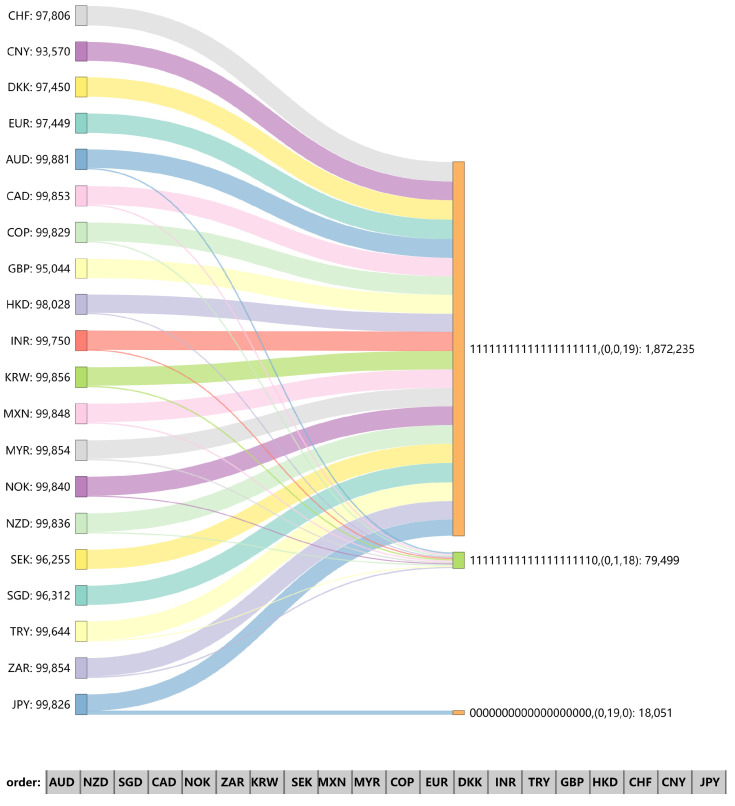
Distribution of final node values in the signed network under different initial node changes during 2010–2019.

**Table 1 entropy-26-00161-t001:** Currency alphabetic code.

Currency	Alphabetic Code	Currency	Alphabetic Code
Australian Dollar	AUD	Won	KRW
Canadian Dollar	CAD	Mexican Peso	MXN
Swiss Franc	CHF	Malaysian Ringgit	MYR
Yuan Renminbi	CNY	Norwegian Krone	NOK
Colombian Peso	COP	New Zealand Dollar	NZD
Danish Krone	DKK	Saudi Riyal	SAR
Euro	EUR	Swedish Krona	SEK
Pound Sterling	GBP	Singapore Dollar	SGD
Hong Kong Dollar	HKD	Turkish Lira	TRY
Indian Rupee	INR	South African Rand	ZAR
Yen	JPY		

**Table 2 entropy-26-00161-t002:** Balanced triads in 1990–1999.

Rank	AUD	CHF	DKK	EUR	GBP	NOK	NZD	SEK
➀		1	1	1				
➁			1	1		1		
➂		1	1			1		
➃		1		1		1		
➄			1	1	1			
➅			1	1				1
➆				1	1	1		
➇			1			1		1
➈				1		1		1
➉			1		1	1		

**Table 3 entropy-26-00161-t003:** Balanced triads in 2000–2009.

Rank	AUD	CHF	DKK	EUR	GBP	NOK	NZD	SEK
➀		1	1	1				
➁			1	1				1
➂			1	1		1		
➃		1	1					1
➄		1		1				1
➅		1				1		1
➆			1			1		1
➇				1		1		1
➈		1	1			1		
➉		1		1		1		

**Table 4 entropy-26-00161-t004:** Balanced triads in 2010–2019.

Rank	AUD	CAD	DKK	EUR	KRW	NOK	NZD	SEK	SGD
➀			1	1				1	
➁			1	1		1			
➂				1		1		1	
➃			1			1		1	

## Data Availability

The links to the public data sets used during the study have been indicated in the main text, and the generated data by analysis are available from the corresponding author under reasonable request.

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
