# Peer review of "Weighted Signed Networks Reveal Interactions between US Foreign Exchange Rates"

_entropy, 2024, doi:10.3390/e26020161_

Round 1

Reviewer 1 Report

Comments and Suggestions for Authors

The authors conduct a study of exchange-rate data over three decades to understand the connections between different currency markets. They construct a measure that allows correlations between fluctuations in different markets to be quantified which can be represented as a network. Properties of these networks are then used to provide a picture of interconnectedness of the markets over time, and to make predictions for how a fluctuation in one market will play out in others. Overall they find that the markets become more interconnected over time, consistent with a picture of increasing globalisation.

Although the methods are described reasonably clearly, I found this paper problematic due to weak conceptual framing. There is little motivation given for the study, for example, questions we might have about how the currency markets are related, and how this relates eg to relevant economic theories. The consequence of this is that the results read as a list of descriptive facts about the data, without yielding insight into the origins or consequences of those facts. I further found the length of this list overwhelming due to the absence of principles to organise its components by.

I would strongly recommend that the authors expand the introduction to explain why we would wish to understand the connectedness of currency markets, what features we are looking for, and what this tells us when we find them. They should potentially also consider reducing the results section to those that are most informative with respect to the framing questions.

If the authors were to resubmit this work, I would also recommend the following changes.

line 42, and elsewhere - The authors talk of power-law cross-correlations between series (actually, on ln 42, they say 'cross-relations', but I think mean 'cross-correlations') but don't explain in what sense this is meant. I would appreciate seeing the function defined that is supposed to exhibit a power law, and in which quantity.

§2.1 - The Euro was not in use as a currency before 1999, so it is not clear how that is accounted for when determining its connections to other currencies in the decade 1990-99

Eq (2) - As written, this allows an arbitrary split of a time series into the trend, seasonal part and residual. Presumably there are some constraints which unambiguously determine each of these terms. These should be explained.

Line 146 - It seems that the authors have a specific technical meaning of 'trend' in mind, which should be explained.

Fig 1 - My understanding is that s is a window over which running averages are computed. I don't understand the rationale for choosing s so as to maximise the correlation coefficients. Can the reasoning / intuition behind this choice be explained?

Fig 2 - Due to a colour vision deficiency, I am not able to distinguish the red and green lines in this figure. Given the small number of red lines reported, I think probably all readers would benefit from the negative correlations being more prominent and distinguishable from the very many positive ones. It is also confusing that different currencies appear in different positions around the periphery as time progresses. This makes comparing the decades very difficult.

Fig 3 - I found it very difficult to make sense of the different centrality measures. It is curious that BC seems to anti-correlate with the other measures. Is there an explanation for this? The lack of a consistent variation in the centrality measures from one currency to the next makes me wonder if any particular measure should be preferred, and if so why. Otherwise it seems multiple interpretations of the data are possible. The text accompanying this figure was very difficult to follow, being essentially a narrative description of the figure, rather than offering a robust interpretation of it (see main point).

Comments on the Quality of English Language

The writing is basically fine in terms of grammar etc, but overall needs to be more precise. Greater focus on the main messages, as described in the main point, will help convey the main ideas of the paper more simply.

Reviewer 2 Report

Comments and Suggestions for Authors

Weighted signed networks reveal interactions between US foreign exchange rates”, Entropy 2839016

This paper studies exchange rates among 21 currencies.  The paper eliminates currencies without data from 1990 to 2019, but it is unclear why.  The data are pre-screened for time trends and seasonal effects, among other things, so why does it matter whether the data cover the same total period?

Line 466: “our method has the potential to predict the fluctuation of the exchange rate market.”  Possibly, but all of the intense analysis here is within the sample period, so internal validity is very high while external validity is completely unexamined.  Can, in fact, this analysis predict anything after 2019 accurately?  The claim to be able to predict should be deleted or tested, as in other macroeconomic and international monetary research.

In Figure 1, 29, 56, and 82 are very different values.  How can the average, 53, be justified?

What does this paper have to do with physics?  This is not a new contribution to statistics, as the methods are already published as the authors indicate.  This is an analysis of exchange rates.  The word “physics” does not appear in the paper.  The submission of this paper to the Statistical Physics section of Entropy is inexplicable.  Obviously, this paper should be in a macroeconomic or monetary economics journal.

Some minor editing:

“Exchange ratio” one time

Equation (13):  define how many nearest neighbors.

Line 390:  why “?”

Line 437:  fo

Comments on the Quality of English Language

Minor problems only.

Round 2

Reviewer 2 Report

Comments and Suggestions for Authors

The journal accepted this for revision, so I assume that Statistical Physics is agreeable to studying exchange rates.  The fact that the statistical methods are from physics is clear in the references.